# PbS QD-Coated Si Micro-Hole Array/Graphene vdW Schottky Near-Infrared Photodiode for PPG Heart Rate Measurement

**DOI:** 10.3390/s23167214

**Published:** 2023-08-16

**Authors:** Mingyuan Xu, Yinghao Cui, Tao Zhang, Mengxue Lu, Yongqiang Yu

**Affiliations:** 1School of Advanced Manufacturing, Nanchang University, Nanchang 330031, China; xumingyuan@ncdx.wecom.work; 2School of Microelectronics, Micro Electromechanical System Research Center of Engineering and Technology of Anhui Province, Hefei University of Technology, Hefei 230009, China; 2021111209@mail.hfut.edu.cn (Y.C.); 2020171318@mail.hfut.edu.cn (T.Z.); 2021111202@mail.hfut.edu.cn (M.L.)

**Keywords:** near-infrared photodetectors, van der Waals Schottky photodiode, Si micro-holes array, photoplethysmography, PbS QDs

## Abstract

Near-infrared (NIR) photodetectors (PDs) have attracted much attention for use in noninvasive medical diagnosis and treatments. In particular, self-filtered NIR PDs are in high demand for a wide range of biomedical applications due to their ability for wavelength discrimination. In this work, we designed and then fabricated a Si micro-hole array/Graphene (Si MHA/Gr) van der Waals (vdW) Schottky NIR photodiode using a PbS quantum dot (QD) coating. The device exhibited a unique self-filtered NIR response with a responsivity of 0.7 A/W at −1 V and a response speed of 61 μs, which is higher than that seen without PbS QD coating and even in most previous Si/Gr Schottky photodiodes. The light trapping of the Si MHA and the PbS QD coating could be attributed to the high responsivity of the vdW photodiode. Furthermore, the presented NIR photodiode could also be integrated in photoplethysmography (PPG) for real-time heart rate (HR) monitoring. The extracted HR was in good accord with the values measured with the patient monitor—determined by analyzing the Fourier transform of the stable and reliable fingertip PPG waveform—suggesting its potential for practical applications.

## 1. Introduction

Schottky photodiodes are key detection devices due to their easy and inexpensive fabrication process and simple device architecture [1,2]. Among these devices, Si-based Schottky photodiodes have drawn particular attention as they present an excellent platform for read-out circuits and large-scale photonic integration [3]. Currently, novel semi-metal two-dimensional materials (2DMs) are being used intensively to construct 2DMs/Si van der Waals (vdW) Schottky photodiodes, owing to their unique advantages such as their dangling bonds and high transmittance [4,5]. Interestingly, high barrier heights and high responsivity can be achieved from graphene (Gr)/Si and MXene/Si vdW Schottky photodiodes, which can be used to create high-performance and low-cost Si-based broadband photodetectors (PDs) [6,7,8]. As is well known, Si-based photodiodes have been used as key components for devices with applications in near-infrared (NIR) light detection. Nowadays, NIR PDs are attracting much attention in the field of noninvasive medical diagnosis and treatment, including specific molecular detection, imaging, biological detection, and light-based diagnostics and therapeutics [9,10,11]. In particular, self-filtered or narrow-band Si-based NIR PDs are in high demand for a wide range of biomedical applications due to their strong ability for wavelength discrimination. For example, a self-filtered NIR Si-based photodiode has been used as a photoplethysmography (PPG) sensor for heart rate (HR) monitoring. Notably, a more accurate HR has been with an Si-based NIR PD than that obtained by using broadband PDs from a commercial PPG system, due to the suppression of disturbances caused by ambient light [12]. Due to the need for noninvasive, inexpensive, and convenient health monitoring, PPGs have become a powerful platform for measuring physiological parameters in individuals using wearable electronics. An individual’s HR, blood pressure (BP), and oxygen saturation (SpO_2_) can be extracted by processing the PPG signals, which contain information on arterial blood volume and cardiac, vascular, and respiratory status [13,14,15]. NIR light—especially in the range of 800–1000 nm—has a higher penetration depth into tissue than visible light, enabling physiological signals to be detected across a wider range [16]. Typically, Si-based NIR PDs can be fabricated to meet the prerequisites for NIR response detection by using complex optical systems—inevitably hindering their integration into PPG sensors. Developing a self-filtered Si-based vdW Schottky photodiode with high-performance is imperative and still remains a challenge.

In this paper, we designed and then fabricated a Si micro-hole array/graphene (Si MHA/Gr) van der Waals (vdW) Schottky photodiode using PbS QD-coating. The Si MHA and PbS QDs were used to improve the photodiode’s NIR absorption and photoresponse. The PbS QD-coated vdW Schottky photodiode possessed a self-filtered NIR response with a responsivity of 0.7 A/W at −1 V and a response speed of 61 μs, which is higher than most previous 2DMs/Si vdW Schottky photodiodes in the range of NIR light. Furthermore, stable and reliable fingertip photoplethysmogram (PPG) signals were achieved based on our self-setup PPG system by using the PbS QD-coated vdW Si MHA/Gr vdW Schottky photodiode. The further extracted HR based on the PPG signals was in good accord with patient monitoring, as determined by the analysis of the Fourier transform of the PPG waveform. Compared with previous Si-based NIR PDs, the self-filtered Si-based NIR photodiode possesses simple methods for device fabrication at a low cost, avoiding complex optical systems and integrating into PPG sensors easily. The self-filtered NIR photodiode shows a potential in PPG systems for achieving more accurate PPG waveforms than those achieved with previous broadband PDs of commercial PPG systems in the presence of ambient light disturbances. 

## 2. Materials and Methods

TCAD simulations. The Silvaco TCAD was used for the design of the silicon micro-hole-based NIR Schottky photodiode. The simulation and analysis of the devices were carried out using this software (Altas, Tonyplot). 

Device fabrication. The PbS-coated Si MHA/Gr vdW Schottky photodiode fabrication process is presented in Figure 1a. A double-side polished silicon wafer (*n*-type, 0.01–1 Ωcm^−1^) was chosen. The Si MHA (diameter~10 μm, spacing~20 μm) was firstly defined on one side of the silicon substrate using UV lithography, and was then etched using an inductively coupled plasma system (ICP, ICP-601) with a HBr/Cl_2_/O_2_ (100/50/50 sccm) mixture for 30 min. The chamber pressure was kept at 10 Pa and the inductive power and chuck bias power were maintained at 120 W and 100 W, respectively. Afterwards, the remaining photoresistant mark was removed by using a cleaning stripper. Subsequently, the chemistry vapor deposition (CVD)-grown graphene bilayer (Hefei Vigon Material Technology Co., Ltd., Hefei, China) was transferred to the other side of the silicon substrate to act as the Schottky contact electrode. A hollow, circular In/Ga bilayer paste electrode was formed by after-etching onto the micro-hole array, serving as an ohmic contact electrode and the light illumination window (diameter of ~5 mm). PbS QDs were synthesized by following previously reported work [17,18,19], and the PbS QD film was then formed on the top of the Si MHA using the spin-coating method. The device was then mounted onto a printed circuit board for the following measurements. The height of the micro-holes was measured by a step profiler (Dektak XT, BRUKER, Mannheim, Germany). The absorption spectra of the device were measured by a UV–vis–NIR spectrophotometer (UV-3600, SHIMADZU, Kyoto, Japan). The images were obtained using a field-emission scanning electron microscope (SEM, SU8020, Hitachi, Kyoto, Japan). 

*Device characterization.* The electrical and optical characteristics of the vdW Schottky photodiodes were measured using a semiconductor analyzer (4200-SCS, Keithley, Cleveland, OH, USA). The laser diodes (254 nm, 365 nm, 405 nm, 650 nm, 808 nm, 980 nm, 1064 nm) were used as light sources to obtain the responsivity of the devices. The incident power was defined on the detector’s active area and calibrated using a Thorlabs PM100D optical power meter. 

*PPG heart rate measurement.* The PPG heart rate measurement was carried out using the PbS QD-coated Si MHA/Gr vdW Schottky photodiode, an amplifier, an LED (980 nm), and a digital oscilloscope (GDS-1072B, RIGOL, Beijing, China), The heartbeat waveform of the same person was simultaneously recorded by a digital oscilloscope from this heart rate measurement system and a patient monitor (UT4000Apro, GOLDWAY, Shenzhen, China).

## 3. Results and Discussion

Figure 1a illustrates the fabrication process of the PbS QD-coated Si MHA/Gr vdW Schottky NIR photodiode, in which the inductively coupled plasma (ICP) technique and the well-developed silicon etching technique were chosen to obtain the Si MHA [8,12]. The CVD-graphene was used to form a Schottky contact as the bottom electrode. The top In/Ga paste electrode was made all around the silicon surface to form good ohmic contact with the n-type silicon, together to keep a high transparency. The photodiode was then mounted onto a circuit board for photoresponse characteristics evaluation and following practical applications in PPG heart rate monitoring. Figure 1b shows the ICP-etched Si MHA image, indicating a good uniformity of the micro-hole array with a diameter of 15 μm and a depth of 2.5 μm (Figure 1c). The diameter variation of the micro-hole can be attributed to lateral etching on the silicon surface due to the longer etching time. From the transmission electron microscope (TEM) image of the PbS QDs (Figure 1d), a uniform QD with a diameter of ~12 nm could be found. Besides this, the absorption of the PbS QDs shown in Figure 1e indicates excellent NIR absorption, which can be used to enhance the NIR response in Si-based photodetectors. 

Figure 2a shows the band energy diagram of the Si MHA/Gr NIR vdW Schottky photodiode; the electron-hole carriers were generated near the junction and then separated, and thereby collected by the electrode under NIR light illumination. Figure 2b shows the *I–V* curves of the photodiode in the dark, showing a rectification behavior. Detailed characterization reveals a Schottky barrier height (*Φ*_B_) of 0.50 eV and an ideal factor (n) of 1.9, which are helpful for separating photo-generated carriers. In order to investigate the photoresponse characteristics, Figure 2c plots the *I–V* curves of the photodiode measured under UV, visible, and NIR light illumination, with the same incident light power (P_in_) of 1.7 mW. Notably, the photodiode showed a sensitivity to NIR light and no obvious response to UV and visible light, which is different from previous Si-based Schottky photodiodes with a broadband response [20,21,22]. Significantly, from the normalized response (Figure 2d), the photodiode exhibited a self-filtered NIR response with a signal to noise ratio (SNR) in the range of 10^2^–10^3^ for visible light and UV light, indicating a strong ability for wavelength discrimination. The unique self-filtered NIR response could attributed to the device structure of the back illuminated Schottky junction, which will be discussed in the following section.

To investigate the enhanced photoresponse characteristics of the PbS QD coating, the *I–V* curves of these photodiodes with and without PbS QD coating under 1064 nm light illumination are shown in Figure 3a, respectively. It was clear that the photocurrent (I_P_~0.1 mA) of the photodiode with the PbS QD coating was higher than that of photodiode without the PbS QD coating (~0.028 mA) at a bias voltage of −2 V. To further determine the device’s performance, Figure 3b plots the *I–V* curves of the photodiode under 1064 nm incident light illumination with power in the range of 0.017 mW to 2.04 mW, suggesting a photoresponse related to the incident light power. Apparently, the photodiode showed photovoltaic behavior, indicating that the photodiode can act as a self-powered photodetector. Besides this, the time-dependent response curves for these photodiodes with and without PbS QD coating at a bias voltage of −1 V under 1064 nm light illumination with different light powers are plotted in Figure 3c, showing a stability current ON/OFF ratio (I_ON_/I_OFF_) of up to 10^2^. Besides this, we calculated the light power-dependent responsivity (*R*), as shown in Figure 3d, which can be expressed as:*R* = I_p_/P_in_(1)

It is apparent that the *R* increases with P_in_ decreases, and reaches up to 0.72 A/W and 0.2 A/W at a bias voltage of −1 V and 0 V, respectively—showing a remarkable enhancement of about two times when compared with the values of the photodiode without the PbS QD coating. The PbS QD-coated Si MHA/Gr vdW photodiode exhibited a higher responsivity for NIR light than previous Gr/Si Schottky photodiodes and even commercial Si-based photodiodes [23,24,25]. In particular, the device could be operated in self-powered mode due to its high responsivity and high I_ON_/I_OFF_ of ~10^5^, which can be seen in Figure 3b.

The enhanced responsivity and self-filtered NIR response of the PbS QD-coated Si MHA/Gr vdW Schottky photodiode were related to be the structure of the Si MHA and the PbS QD coating. Figure 4a compares the incident light (λ)-dependent *R* curves of the planar Si/Gr, Si MHA/Gr, and PbS QD-coated Si MHA/Gr photodiodes under a light power of 0.017 mW and a bias voltage of 0 V. It can be clearly seen that the calculated *R* of the PbS QD-coated Si MHA/Gr photodiode showed improvements of about 6.7 and 2.8 times for 1064 nm light and about 11.8 and 2.0 times for 980 nm light by comparing it with those of the planar Si/Gr and Si MHA/Gr, respectively. Therefore, both Si MHA and PbS QDs can be beneficial in improving NIR responses based on our study, which can attributed to the improvement of NIR light absorption—as shown in Figure 4b. The TCAD-simulated optical intensity distribution was used to understand the mechanism for this, indicating an obvious light trapping effect with the Si MHA (Figure 4c). Furthermore, the absorption of the PbS QDs in NIR can also be beneficial for NIR responses (Figure 1e). Other than this, we can also explore the mechanisms for the self-filtered NIR response of photodiodes based on the simulated optical intensity distribution. From Figure 4c, we can see that long-wavelength NIR light (e.g., 980, 1064 nm) can penetrate to the bottom Schottky junction, while visible light (e.g., 520, 650 nm) will be absorbed by the Si surface due to the absorption coefficient (α) of the Si [26]. When NIR light is shone onto the photodiode, photocarriers can be generated near the junction and then separated and collected by the built-in Schottky electric field—thus inducing a photocurrent. Due to the visible light being absorbed near the Si top surface, most of the photo-carriers will be recombined because of their diffusion length [27], resulting in an extremely low photocurrent.

Finally, we explicitly investigated the optical bandwidth and response speed of the photodiode. The normalized response characteristics versus the pulsed light frequency (*f*) curve of the photodiode at a zero bias voltage is shown in Figure 5a. The 3 dB optical bandwidth was estimated to be 2.0 kHz, which is comparable to most reported Gr/Si photodiodes [28]. Notably, the photodiode was able to follow the pulse light at *f* = 2 kHz well (Figure 5b), and even *f* = 8 kHz (Figure 5c)—suggesting that the photodiode can follow light with high-pulsed frequencies well. From the magnified plot for a single photovoltage cycle (Figure 5d) at *f* = 2 kHz, the rise time (τ_r_) and fall time (τ_f_) were estimated to be 61 μs and 206 μs, which are also comparable to most oreported Gr/Si or Si-based heterojunction photodiodes [28].

The high-performance and self-filtered NIR response of the PbS-coated Si MHA/Gr vdW Schottky NIR photodiode showed great potential for biomedical measurements. We built a transmitted PPG heart pulse monitoring system that was composed of a 980 nm LED, an amplifying circuit, and an oscilloscope—as shown in Figure 6a. In this system, the 980 nm LED was used as the light source because the transmission of tissue is the greatest in the range of 800–1000 nm. The PbS-coated Si MHA/Gr vdW Schottky NIR photodiode was used as the sensor to detect the NIR light signal of the transmitted tough finger tissue. Vascular blood flow induced the transmitted light changes, which could be detected by the vdW Schottky photodiode, and then the amplified photovoltages were simultaneously recorded by an oscilloscope. Figure 6b shows the heart pulse waveform from our heart pulse monitoring system. It was found that a PPG waveform was successfully achieved in our measurements. In particular, the waveforms possessed featured peaks—such as systolic and diastolic peaks (Figure 6c)—suggesting a reliable PPG waveform that is beneficial for measuring subsequent physiological parameters of individuals such as HR and BP. To accurately estimate the value of the heart rate, a PPG waveform was transformed by Fourier transform, as shown in Figure 6d. The HR was estimated to be 87 beats per minute (BPM), which is in good accord with the values measured by the commercial patient monitor (inset in Figure 6d). These results suggest that the Pb-coated Si MHA/Gr vdW Schottky NIR photodiode shows a potential for application in PPG systems to obtain reliable PPG waveforms for heart rate measurements.

## 4. Conclusions

In summary, a PbS-coated Si MHA/Gr vdW Schottky NIR photodiode was successfully fabricated. Compared with previous Si-based Schottky photodiodes, the device exhibited a unique self-filtered NIR response with a responsivity of 0.7 A/W at −1 V and a response speed of 61 μs, which is higher than that achieved without PbS QD-coating and even most previous Si/Gr photodiodes. The device structure, the Si MHA, and the PbS QD coating could be seen to contribute to the NIR response. The presented photodiode was used as a PPG sensor, and a reliable PPG waveform with features could be obtained. The extracted HR based on an analysis of the Fourier transform of the PPG waveform was in good accord with the values measured by the patient monitor, suggesting its potential for practical applications in real-time heart rate (HR) monitoring. Our work opens a new avenue for the fabrication of high-performance NIR Schottky photodiodes for PPG heart rate monitoring.

## 5. Patents

This work was supported by the Open Research Fund of the State Key Laboratory of Pulsed Power Laser Technology (No. SKL2019KF09).

## Figures and Tables

**Figure 1 sensors-23-07214-f001:**
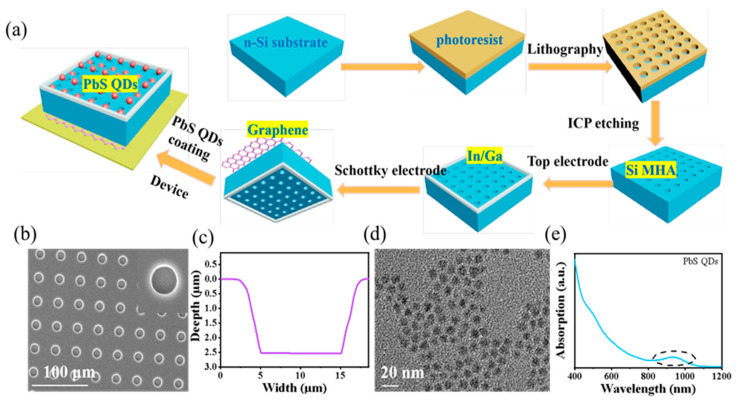
(**a**) Schematic illustration of the fabrication process for PbS QD-coated Si MHA/Gr vdW Schottky NIR photodiode. (**b**) SEM image of the Si MHA. (**c**) The depth curve of the Si MHA. (**d**) TEM image of the PbS QDs. (**e**) The absorption of the PbS QDs.

**Figure 2 sensors-23-07214-f002:**
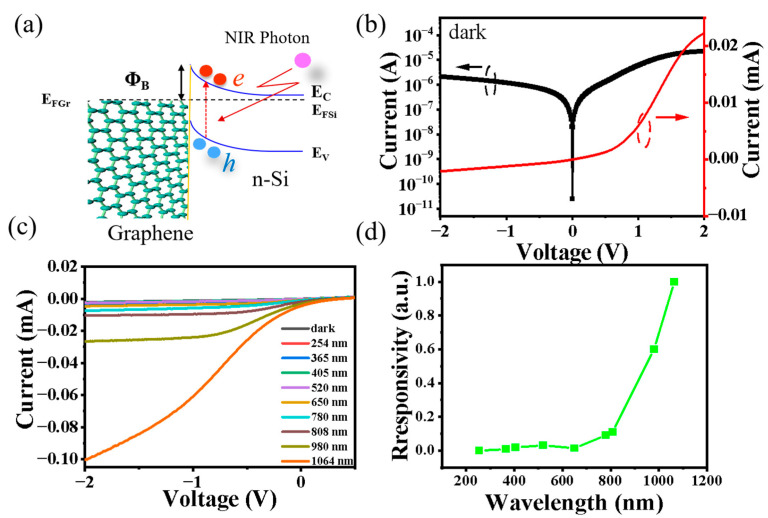
(**a**) The band energy diagram of the Si MHA/Gr NIR vdW Schottky photodiode. (**b**) *I–V* curves of the photodiode in the dark. (**c**) *I–V* curves of the photodiode in the dark and under different types of light illumination at a light power of 1.7 mW, respectively. (**d**) The normalized response.

**Figure 3 sensors-23-07214-f003:**
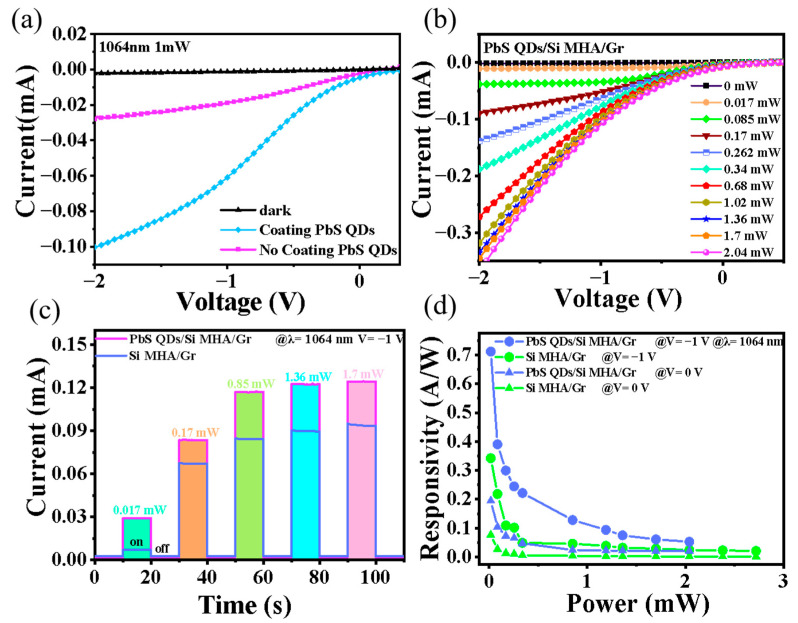
(**a**) *I–V* curves of the vdW Schottky photodiodes with and without PbS QD coating and in the dark and under 1064 nm light illumination, respectively. (**b**) *I–V* curves of the photodiode with PbS QD coating under 1064 nm light illumination with varied light power. (**c**) The time response of the photodiodes with and without PbS QD coating under varied light power illumination. (**d**) The light power dependent responsivity curves of the photodiodes with and without PbS QD coating at a bias of 0 V and −1 V, respectively.

**Figure 4 sensors-23-07214-f004:**
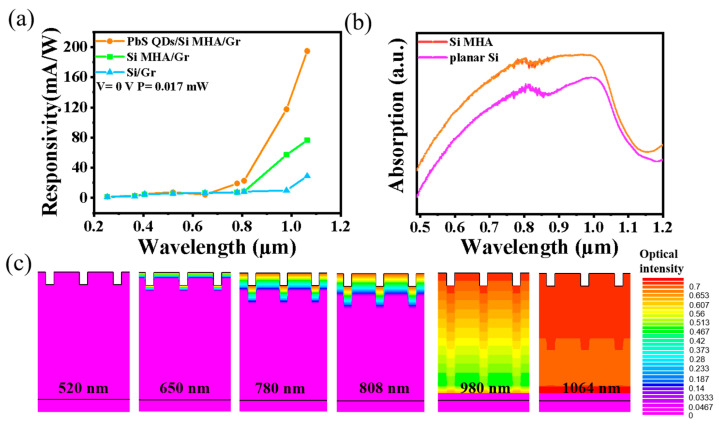
(**a**) The wavelength-dependent responsivity for planar Si/Gr, Si MHA/Gr, and PbS QD-coated Si MHA/Gr photodiodes, respectively. (**b**) The absorption of Si MHA and planar Si. (**c**) The simulated optical intensity distribution of the Si MHA/Gr and planar Si/Gr Schottky junction under different light illumination, respectively.

**Figure 5 sensors-23-07214-f005:**
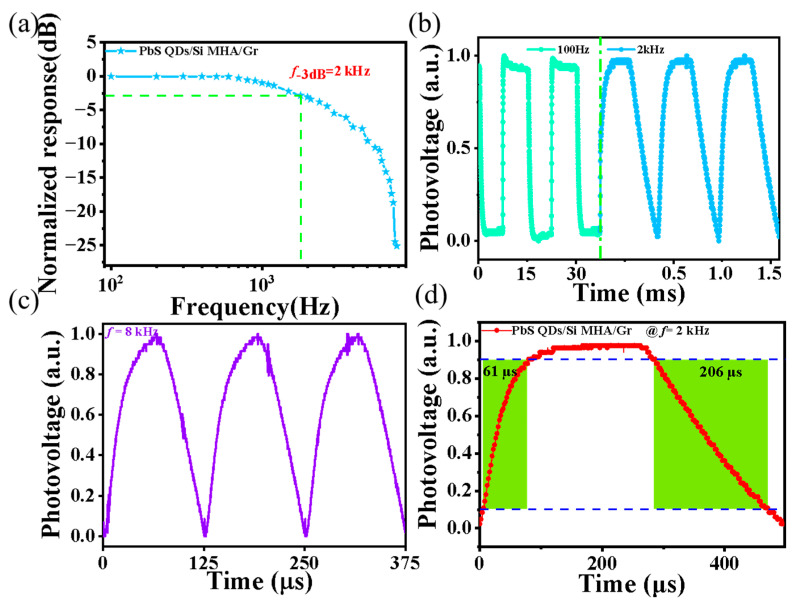
(**a**) Normalized response characteristics versus frequency (*f*) curve of the PbS-coated Si MHA/Gr vdW Schottky photodiode at zero bias voltage. Normalized time response of these photodiodes at *f* = 100 Hz and 2 kHz (**b**) and (**c**) 8 kHz. (**d**) Magnified plot of one response cycle of the photodiode for estimating the response time.

**Figure 6 sensors-23-07214-f006:**
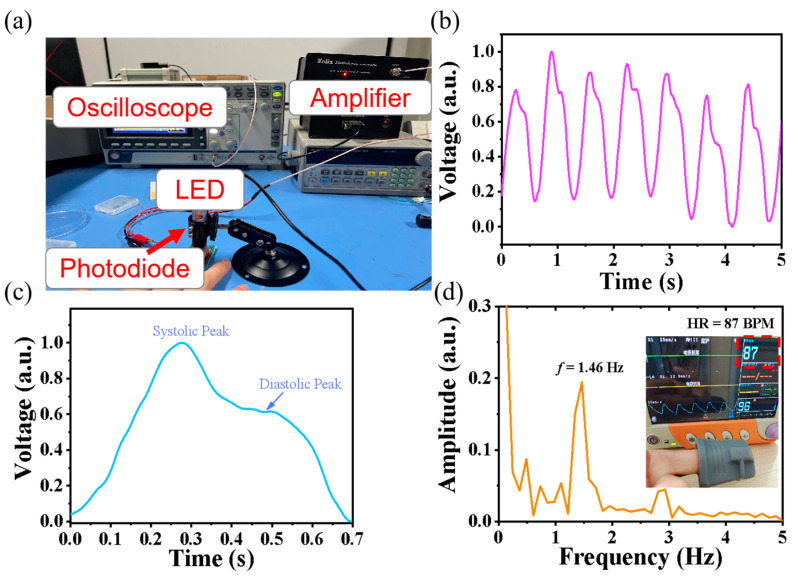
(**a**) Schematic illustration and photograph of the PbS-coated Si MHA/Gr vdW Schottky photodiode for PPG measurement. (**b**) The obtained PPG waveform. (**c**) A typical PPG waveform from (**b**), showing systolic peaks and diastolic peaks. (**d**) Frequency-domain representation of the PPG signal for estimating heart rate. Inset shows the HR measured by a patient monitor.

## Data Availability

The data presented in this study are available on request from the corresponding author.

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
