# Peer review of "PbS QD-Coated Si Micro-Hole Array/Graphene vdW Schottky Near-Infrared Photodiode for PPG Heart Rate Measurement"

_sensors, 2023, doi:10.3390/s23167214_

Round 1

Reviewer 1 Report

This is a well-written manuscript, which provides an original approach for creating a photodiode based on graphene/microstructured Si Schottky junction and demonstrate its application for photoplethysmography measurements. Authors also demonstrate that the performance of their device can be significantly improved by its coating with PbS quantum dots due to increased absorption of light in NIR region.  The article is well written, and the results reported seems correct. Despite all its advantages, the article contains a number of inaccuracies. For example, in Figure 4a the authors mixed up the notations of the spectral dependences of responsivity for PbS/Si MHA/Gr and Si MHA/Gr. In addition, the wavelength of 1064 nm is wrongly indicated in the figure 4a, while the figure represents the spectral dependence of responsivity. It should be corrected. However, these shortcomings do not reduce the overall high level of the manuscript. The simplicity of fabrication and unique self-filtered NIR response make this device very promising for practical applications. Thus, this work is useful for scientific the community.

Additional comments

1) What is the main question addressed by the research?

The main question addressed by this research is following : what is the mechanism of improving the NIR performance of photodiode based on Si/graphene Schottky junction by application of microstructuring and modifying the architecture by adding the PbS QDs and can this device be used for applications in photoplethysmography?

2) Do you consider the topic original or relevant in the field? Does it address a specific gap in the field?

Yes, the topic seems very original since the self-filtered NIR photodiodes are in high demand for many photonics applications, but common solutions are usually quite difficult to fabricate or they lack the desired characteristics.

3) What does it add to the subject area compared with other published material?

This study adds the understanding of the mechanisms of light absorption and photoexcitation of carriers in such complex structure as PbS coated Si MHA/Gr vdW Schottky photodiode.

4) What specific improvements should the authors consider regarding the methodology? What further controls should be considered?

The methodology can be improved if authors address the theoretical or experimental investigations of carriers’ photoexcitation and separation on several heterojunctions as well as light absorption by microstructured silicon surface.

4) Are the conclusions consistent with the evidence and arguments presented and do they address the main question posed?

The results are consistent with the evidence and arguments presented and they fully address the main question posed as soon as in conclusions the authors demonstrate the improvement of the photoresponsivity of PbS coated Si MHA/Gr vdW Schottky photodiode in comparison with previously studied Si/Gr photodiodes. Moreover, the authors demonstrate that their device is capable of being used as a key component of heart rate monitoring system.

5) Are the references appropriate?

The references used in this article are quite appropriate, the authors cite novel articles in high-impact journals and all the citations are referent with the subject of study.

6) Please include any additional comments on the tables and figures.

It is already commented that the Fig. 4 should be corrected. There are no more additional comments.

The text contains a small number of typos. Thus, line 194 ...most of the photocarriers will be recombines

Reviewer 2 Report

The paper describes the construction characterization of a Schottky photodiode incorporating  PbS QDs coated Si micro-holes array/Graphene vdW Schottky

and a test of the near-infrared photodiode for a PPG heart-rate measurement

Notably, the photodiode shows an sensitivity to NIR not UV making it different from Si-based Schottky photodiodes also the photodiodes show photovoltaic behavior, indicating that they can be self powered.

The potential for incorporating into a real-time heart-rate (HR) monitor is demonstrated

The general format and results are good, but the English must be improved and there are several problems with the figures and how they are described

Explain self filtering in the introduction, how is the current device better than existing ones in heart rate monitors, are they going to be easier/cheaper to manufactur?

What is good about having a higher response time doesn’t that mean a longer response time, perhaos the authors mean better?

Figure 3 (a)how do the colours in the plot relate to the legend I cant discern the colour differences, symbols are too small??

Figure 3 C what does the green indicate,  how are the different light powers indicated?

Figure 4 What does c indicate and how?

Figure 4 d does not appear to exist?

Line 179 -181

"It can be clearly seen that the calculated R of PbS QDs coated Si MHA/Gr photodiode shows an improvement about 9.5 and 2.8 times by comparing with those of planar Si/Gr and Si MHA/Gr, respectively."

 does not appear to represent what is in Figure 4 a, the ratios are 2.8 and 9.5 times not as indicated in the text.

The spelling is generally OK although photodioode  in line 111 is not a word.  Please pay attention to the grammar, preferably with the help of  a native speaker or some good software

Round 2

Reviewer 2 Report

The authors response and everything looks in order except for one comment. Figure 3a is OK now but I notice that 3b has the same problem. I should have pointed out before, it is still not clear there are colors or symbols on the legend which do not appear on the graph that the symbols were small which makes the line and the symbols indistinguishable, and since you only see one color for each series, you are either seeing the symbol or the line but it isn't exactly clear then what series is what.

-
